# Overexpression of TRIM16 Reduces the Titer of H5N1 Highly Pathogenic Avian Influenza Virus and Promotes the Expression of Antioxidant Genes through Regulating the SQSTM1-NRF2-KEAP1 Axis

**DOI:** 10.3390/v15020391

**Published:** 2023-01-30

**Authors:** Yanwei Liu, Yifan Wei, Ziwei Zhou, Yongxia Gu, Zifeng Pang, Ming Liao, Hailiang Sun

**Affiliations:** 1College of Veterinary Medicine, South China Agricultural University, Guangzhou 510642, China; 2Key Laboratory of Zoonosis Control and Prevention of Guangdong Province, South China Agricultural University, Guangzhou 510642, China; 3National and Regional Joint Engineering Laboratory for Medicament of Zoonosis Prevention and Control, South China Agricultural University, Guangzhou 510642, China

**Keywords:** TRIM16, NRF2, AIV, ubiquitination, oxidative stress

## Abstract

Oxidative stress plays a vital role in viral replication. Tripartite motif containing 16 (TRIM16) is involved in diverse cellular processes. However, the role of TRIM16 in oxidative stress induced by infection of the highly pathogenic H5N1 avian influenza virus (HPAIV) is unclear. We found that under conditions of H5N1 HPAIV infection, reactive oxygen species (ROS) levels in A549 cells peaked at 24 h post infection (hpi), and antioxidant genes’ expression levels were down-regulated. Overexpression of TRIM16 in A549 cells resulted in a decrease in the titter of H5N1 HPAIV and led to significant up-regulation of the antioxidant genes’ expression levels, which indicates that TRIM16 positively regulates the sequestosome 1/Kelch-like associated enoyl-CoA hydratase 1 protein/nuclear factor erythrocyte 2-derived 2-like 2 (SQSTM1/NRF2/KEAP1) pathway. Under basal conditions, TRIM16 led to a modification of NRF2 through an increase in K63-linked poly-ubiquitination of NRF2. Collectively, our findings provide new insight into understanding TRIM16′s role in anti-oxidative stress in H5N1 HPAIV infected A549 cells.

## 1. Introduction

Avian influenza virus (AIVs) is classified into highly pathogenic avian influenza viruses (HPAIV) and low pathogenic avian influenza viruses (LPAIV) according to their pathogenicity to chickens [1]. In 1997, the H5N1 avian influenza virus killed 6 of the 18 infected people in Hong Kong. This event was the first time that a deadly H5N1 virus had been reported in humans, which attracted widespread attention [2]. Since 2003, 865 human cases of H5N1 viral infections have been reported in more than 20 countries. (Data were obtained from the World Health Organization (WHO) website.) H5N1 HPAIV causes huge economic losses in the poultry industry and poses serious threats to public health [3].

Cells use oxygen as the final electron acceptor for mitochondrial energy metabolism; thus, cells inevitably produce oxygen-containing byproducts. Because the resulting by-products tend to be highly reactive, they are often referred to as “reactive oxygen species” (ROS). These ROS mainly include superoxide (O^2−^), hydrogen peroxide (H_2_O_2_), hydroxyl radicals (OH^−^), and singlet oxygen [4]. Under normal cellular metabolism, ROS are continuously being produced. Once the accumulation of ROS exceeds the clearance of oxides, the balance between the oxidative and antioxidant systems is disrupted, resulting in an increase in neutrophil inflammatory infiltration, increase in protease secretion, and accumulation of oxidative intermediates, ultimately leading to tissue damage, inflammatory responses, and apoptosis. Pathogenic microbial infection has also been shown to be a strong trigger for oxidative stress, thus causing disruption in the oxidative and reduction states of host cells and thus favoring the survival of pathogenic microorganisms [5,6,7]. Some pathogenic microorganisms, such as Dengue (DENV) [8] and human immunodeficiency virus (HIV) [9], induce large quantities of ROS in cells during infection, which is conducive to their survival in the host. Cellular infection with DENV results in an increase in ROS levels within cells and enhancement of viral replication in host cells [8]. HIV-induced oxidative stress promotes viral survival in astrocytes [9].

Nuclear factor erythroid 2-related factor 2 (NRF2) is a key transcription factor that plays an integral and important role in combating various stress responses including oxidative stress, protein toxic stress, and electrophilic insults [10,11,12]. Kelch-like enoyl-CoA hydratase (ECH)-associated protein 1 (KEAP1) interacts with NRF2 and mediates NRF2 degradation under basal cellular conditions [11]. If cells are exposed to oxidative stress and electrophilic damage, KEAP1 loses its original activity leading to an NRF2 nuclear shift followed by transcription inducing the expression of cytoprotective and antioxidant genes [13]. Subsequently, the expression of antioxidant response elements (ARE), such as heme oxygenate 1 (Hmox1) and quinone dehydrogenase 1 (NQO1), increased due to the shift of NRF2 to the nucleus. Specific natural and synthetic compounds can down-regulate DNMTs and/or HDACs inhibiting the methylation status of NRF2 promoter, then reactivating the expression of NRF2 protecting normal prostatic cells from ROS damage and tumorigenesis [14]. NRF2 activation in cancer cells is responsible for the development of chemoresistance inactivating drug-mediated oxidative stress that normally leads cancer cells to death [15]. The autophagy adaptor protein sequestosome 1 (SQSTM1) has been shown to play a role in the positive regulation of NRF2 by influencing NRF2–KEAP1 interactions or promoting KEAP1 sequestration and autophagy degradation [16,17,18,19]. SQSTM1 stabilizes NRF2 leading to further amplification of the NRF2 response [17]. However, the molecular mechanism of NRF2 alleviating oxidative stress during viral infection remains unclear.

Tripartite motif (TRIM) proteins are a family of about 75 proteins that have different functions during cellular development. They play an important role in innate immunity, cell proliferation, differentiation, development, cancer, apoptosis, and autophagy [20,21]. The control of intracellular signaling in native immune activation has been shown regulated by post-translational modifications such as phosphorylation, acetylation, and ubiquitination [22]. The first studies of the relationship between TRIM protein and innate immunity suggest that TRIM5a prevents infection by several viruses, including lentiviruses and retroviruses, but human TRIM5a cannot interfere with HIV infection [23]. Suppressor of cytokine signaling (SOCS1) binds to TRIM8 (GERP) to stabilize SOCS1 proteins, thereby inhibiting interferon gamma (IFNγ) signaling through the action of SOCS1 [24]. Most members of the TRIM family have a conserved really interacting new gene (RING) and B-box coiled-coil (RBCC) domain-containing factors, and are characterized by a triple motif structure formed by three domains (tripartite motif): (1) the zinc finger domain (RING finger), (2) one or two B-boxes, and (3) one crimp helix domain [25]. Notably, another TRIM subfamily in the TRIM family that contains a deleted RING domain, such as Trim16 (protein 16 with tripartite motifs) was found [25,26]. TRIM16 was first identified as an estrogen-responsive B-box protein (EBBP) that possesses transcriptional activity. Further studies confirmed that TRIM16 plays an important role in the secretion of interleukin-1β and the transcriptional regulation of retinoic acid receptor β2 [27]. Growing evidence that TRIM16 has different functions in cell differentiation, innate immune response, and tumorigenesis has been reported [28,29,30]. Previous studies have shown that TRIM16 is an oxidative stress-inducing gene [31,32,33]. However, the effects of TRIM16 on viral infection have rarely been reported, and it was only found that after infection with HBV virus that the expression of TRIM16 was significantly up-regulated [34]. 

In this study, we investigated the role of TRIM16 in H5N1 HPAIV-induced oxidative stress. First, we determined the specific time at which the H5N1 HPAIV induced oxidative stress in host cells. Next, we studied the effects of TRIM16 in H5N1 HPAIV replication. Finally, we demonstrate the role of TRIM16 in the antioxidant pathway SQSTM1/NRF2/KEAP1. Our findings provide new insight into understanding TRIM16′s role in anti-oxidative and AIV replication in host cells.

## 2. Materials and Methods

### 2.1. Viruses and Cells

The H5N1 subtype avian influenza virus, A/duck/Guangdong/212/2004 (DK/212) was isolated and stored in the laboratory [35]. The virus was propagated in 9–11 days in specific pathogen-free (SPF) chicken embryos. Human lung carcinoma cells (A549), Madin–Darby canine kidney cells (MDCK), and human embryonic kidney cells (HEK293T) were grown in Dulbecco’s modified Eagle’s medium (DMEM, GIBCO, Grand Island, NY, USA) supplemented with 10% fetal bovine serum (FBS, GIBCO, Grand Island, NY, USA), 100 U/mL penicillin, and 100 mg/mL streptomycin. The cells were incubated at 37 °C, 5% (*v*/*v*) CO_2_. The 50% tissue culture infectious dose (TCID_50_) A/duck/Guangdong/212/2004 was calculated in MDCK cells. All experiments involving H5N1 HPAIV were conducted in a biosafety level 3 laboratory (BSL-3).

### 2.2. Construction of Plasmids

The human TRIM16 gene (NM_001348119.1) was obtained from the cDNA of A549 cells by polymerase chain reaction (PCR). The purified PCR product was then cloned into the PCAGGS vector, which contains an HA tag in the N terminal, and FLAG tag in the C terminal and an eGFP and an AsRed tags in the N terminal. The coding sequence of human SQSTM1 (NM_003900.5), NRF2 (NM_006164.5), and KEAP1 (NM_203500.2) from A549 cells with FLAG, HA, and eGFP tags were cloned into the PCAGGS vector to obtain PCAGGS-HA-SQSTM1, PCAGGS-SQSTM1-FLAG, PCAGGS-eGFP-NRF2, PCAGGS-FLAG-NRF2, and PCAGGS-HA-KEAP1 expression recombinant plasmids. All recombinant plasmids were identified by sequencing.

### 2.3. ROS Detection

A549 cells infected with DK/212 and control cells were washed twice with preheated phosphate-buffered saline (PBS). Five micromolar of CM-H2DCFA was diluted in DMEM medium without FBS (Invitrogen, Carlsbad, CA, USA) was added and incubated at 37 °C for 30 min. The cells were then washed twice with PBS and separated using 0.25% trypsin. Fluorescence-activated cell sorting (FACS) analysis was performed using the BD FACSCalibur system. At least 50,000 cells were analyzed, and the average channel fluorescence intensity was calculated.

### 2.4. Reverse Transcription Quantitative Polymerase Chain Reaction (RT-qPCR)

TRIzol (Invitrogen, Carlsbad, CA, USA) was used to extract total RNA from differently treated cells according to standard instructions. The quality and quantity of RNA in the samples were determined by the uitro-spec2000 mass spectrophotometer. One microgram of RNA was reverse transcribed into cDNA using M-ML-V reverse transcriptase (Promega, Madison, WI, USA), Oligo-dT (TaKaRa, Tokyo, Japan) and random primers (TaKaRa). Next, cDNA was used as a template for qPCR detection using Go Taq qPCR Master Mix (Promega, Madison, WI, USA) and SYBR Green I. The reverse transcription quantitative polymerase chain reaction (qPCR) reaction was performed using the CFX96™ real-time system (Bio-Rad, Hercules, CA, USA) under specific cycling conditions: (1) one cycle at 95 °C for 2 min, (2) 40 cycles at 95 °C for 15 s, and (3) 60 °C for 40 s. Primers for RT-qPCR were designed using the OLIGO 7 primer analysis software (Molecular Biology Insights Inc., Cascade, CO, USA) (Table 1). The relative transcription levels of all genes were calculated by 2^−ΔΔCt^ and glyceraldehyde 3-phosphate dehydrogenase (GAPDH) gene was used as the standard. Each data point represents results from three independent experiments.

### 2.5. Co-Immunoprecipitation

HEK 293T cells were cultured to 80% confluence and then transfected with the associated genes with lipo2000 (Invitrogen, Carlsbad, CA, USA). At 24 h post transfection, cells were washed twice with PBS, and cell lysate (Beyotime, Jiangsu, China) containing 1Nm protease inhibitor (PMSF, Dingguo, Beijing, China) was added to cells and incubated at 4 °C for 30 min after which the mixture was centrifuged at 4 °C for 10 min at 12,000 rpm. The supernatant was incubated with 50 Μl agarose beads (Thermo Fisher Scientific, Waltham, MA, USA) and shaken at 4 °C for 6 h. After incubation, the mixture was washed three times with Tween/Tris-buffers saline TBST and boiled in 5× sodium dodecyl sulfate (SDS) loading buffer (Dingguo, Beijing, China) for 10 min.

### 2.6. Western Blotting

Cell lysate (Beyotime, Jiangsu, China) with 1 Nm protease inhibitor (phenylmethylsulfonyl fluoride [PMSF], Dingguo, Beijing, China) was used to prepare cell lysate products, which were lysed on ice for 30 min and then centrifuged for 10 min at 12,000 rpm and 4 °C. Protein concentration was determined with BCA kits (Invitrogen, USA). Loading buffer (5× SDS) was added into the sample and then was boiled for 10 min. Protein lysates were separated on SDS–polyacrylamide gels, transferred onto nitrocellulose membrane (Millipore, Burlington, MA, USA), and blocked in 5% BSA (Dingguo, Beijing, China) solution for 1 h. The nitrocellulose membrane was then washed with TBST three times and incubated with the primary antibody overnight at 4 °C. The secondary antibodies were DyLight 800 goat antimouse or antirabbit IgG (H+L) (Thermo Fisher Scientific, Waltham, MA, USA).

The primary antibodies used in Western blotting with dilutions included TRIM16 (abcam 72129, 1:1000), SQSTM1 (BD #610832; 1:2000), NRF2 (Santa Cruz #SC-365949; 1:1000), KEAP1 (CST #8047S; 1:1000), HA (Thermo Fisher Scientific 20182, 1:2000), FLAG (Thermo Fisher Scientific A36803, 1:2000), and green fluorescent protein (eGFP; Thermo Fisher Scientific F56-6A1.2.3, 1:2000).

### 2.7. Confocal Microscopy

To study the colocalization of TRIM16 and NRF2, HEK 293T cells were inoculated on glass-bottom Petri dishes and co-transfected with green fluorescence PCAGGS-eGFP-NRF2 and red fluorescence pCAGGS-AsRed-TRIM16 using a lipo2000 transfection reagent. After co-transfection of plasmids for 24 h, the cells were washed with PBS twice, fixed with 4% paraformaldehyde for 10 min, and infiltrated with 0.1% Triton X-100 at 4 °C for 10 min. After being washed with PBS three times, the cells were stained with 1 µM of 4′, 6-diaminyle-2-phenyl indole (DAPI) for 5 min. Fluorescence images were obtained using a confocal TCS SP8 microscope (Lecia, Wetzlar, Germany) under a ×100 oil objective lens.

### 2.8. Overexpression of HA-TRIM16 Reduces the Titer of DK/212 Virus

A549 cells were plated in a 6-well plate and cultured to 70% confluence after which 4 μg HA-TRIM16 or empty vector was introduced into the cells using the lipo2000 transfection reagent (Invitrogen, Carlsbad, CA, USA) according to the manufacturer’s instructions. At 24 h after transfection, A549 cells were infected with DK/212 virus at 0.1 multiplicity of infection (MOI). After 1 h of adsorption, cells were washed with PBS twice, and DMEM containing 2% FBS was added for cell culture. The supernatant was collected at 12, 24, 36, and 48 h and titrated in MDCK cells.

### 2.9. Statistical Analysis

Data were expressed as mean ± standard deviation and statistically analyzed using GraphPad Prism 9 Software (GraphPad Software Inc., San Diego, CA, USA). The *t*-test was used for inter-group comparison, and a two-factor analysis of variance was used for inter-group comparison. All *p*-values were considered significant at 0.05, and the difference was statistically significant.

### 2.10. Ethics Statement

The cell infection study was reviewed and approved by the Administration and Ethics Committee of the South China Agricultural University.

## 3. Results

### 3.1. Cells Were under Oxidative Stress after Infection with H5N1 HPAIV

A549 cells were infected with DK/212 virus at 0.1 MOI, and intracellular ROS levels were monitored at 12, 24, and 36 h post infection (hpi). At 24 and 36 hpi, intracellular ROS levels were significantly increased compared to controls, and the quantity of ROS peaked in A549 cells at 24 hpi (Figure 1D–F). Compared to that at 12 hpi, the quantity of ROS in A549 cells showed a sharp increase trend at 24 hpi (Figure 1A–F), and decreased at 36 hpi (Figure 1D–I). The results showed that the DK/212 virus caused a significant increase in ROS levels in A549 cells and exposed cells to a state of intense oxidative stress.

### 3.2. H5N1 HPAIV Inhibits the Expression of Antioxidant Genes

To characterize antioxidant gene expression in A549 cells after infection with the H5N1 subtype influenza virus 24 h, relative quantitative levels of NRF2, HO-1, NADPH, GCLC, GCLM, GPX2, and SLC7A11 gene expression levels were detected using qRT-PCR. Variations in the expression of NRF2, HO-1, NADPH, GCLC, GCLM, GPX2, and SLC7A11 were observed in A549 cells infected with DK/212. Compared to that of the control group, the mRNA expression level of NRF2 at 24 hpi had decreased (*p* < 0.01) as shown in Figure 2A. The protein expression of NRF2 was also significantly down-regulated at 24 hpi in the treatment group (Figure 2H). After the influenza virus infection, the expression of GCLC, GCLM, and SLC7A11 at the mRNA level was significantly down-regulated (*p* < 0.01, *p* < 0.001, and *p* < 0.01, respectively) compared to the control group (Figure 2B–D). Compared to the control group, the expression of NADPH, GPX2, and HO-1 was significantly down-regulated at 24 hpi (*p* < 0.05) as shown in Figure 2E–G. These results showed that the H5N1 subtype influenza viral infection caused significant inhibition of the expression of related antioxidant genes.

### 3.3. Overexpression of HA-TRIM16 Reduces the Titer of DK/212 Virus in A549 Cells

TRIM16 protein played a key role in stabilizing the NRF2 pathway [36]. In order to explore the effect of the H5N1 subtype influenza virus on TRIM16 protein, A549 cells were infected with DK/212 virus and the expression of the TRIM16 protein was detected by Western blot at 24 hpi. The results show that after infection with the H5N1 subtype influenza virus TRIM16 protein show significant down-regulation (Figure 3A). To explore the effect of TRIM16 overexpression on replication of H5N1 subtype avian influenza virus, A549 cells were infected with DK/212 virus at 24 h after transfection PCAGGS-HA-TRIM16 or PCAGGS empty vector. The titers of DK/212 increased gradually from 12 to 48 hpi in cells transfected with HA-TRIM16 or transfected with PCAGGS empty vector. The titers of DK/212 virus were significantly lower in TRIM16-overexpressed cells than that in cells transfected with PCAGGS empty vector at 24 and 36 hpi (*p* < 0.05) (Figure 3B). These results suggest that by overexpressing TRIM16, the titer of the H5N1 subtype avian influenza virus was reduced in A549 cells.

### 3.4. Overexpression of TRIM16 Promotes the Expression of Antioxidant Genes

The effects of TRIM16 on antioxidant genes, the relative quantitative levels of HO-1, NADPH, glutamate–cysteine ligase catalytic subunit (GCLC), glutamate–cysteine ligase modifier subunit (GCLM), glutathione peroxidase 2 (GPX2), and solute carrier family 7 (SLC7A11) gene expression in A549 cells transfected with HA-TRIM16 were examined using qPCR. Compared to the cells transfected with PCAGGS empty vector, the mRNA levels of GCLC and GCLM in the cells overexpressing TRIM16 were significantly up-regulated by 1.88- and 2.16-fold, respectively (*p* < 0.001) (Figure 4A,B). The expression of mRNA levels of HO-1, NADPH, and GPX2 were significantly up-regulated to 1.85-, 1.44-, and 1.33-fold that of the control group, respectively (Figure 4C–E). It is very interesting that the expression level of SLC7A11 mRNA was significantly down-regulated (*p* < 0.0001) as shown in Figure 4F. In summary, overexpression of TRIM16 in A549 cells leads to an increase in the mRNA expression of related antioxidant genes.

### 3.5. Interaction and Colocalization of TRIM16 and NRF2

To determine whether TRIM16 and NRF2 had protein interactions, HA-TRIM16 and FLAG-NRF2 were co-transfected into HEK 293T cells to perform an exogenous co-immunoprecipitation (CoIP) assay. When the lysates were immunoprecipitated with anti-HA agarose beads, FLAG-NRF2 CoIP was transfected with HA-TRIM16 using an anti-FLAG antibody. Therefore, the results of CoIP indicate that protein interaction between TRIM16 and NRF2 had occurred (Figure 5A). To study the colocalization phenomenon of TRIM16 and NRF2, the eGFP gene was cloned into pCAGGS-NRF2 to construct the eGFP-NRF2 plasmid with green fluorescent expression and the AsRed gene was transfected into PCAGGS-TRIM16 to construct AsRed-TRIM16 and construct a red fluorescent expression plasmid. Colocalization of TRIM16 with NRF2 in HEK293T cells was examined using laser confocal microscopy. When eGFP-NRF2 or AsRed-TRIM16 was expressed alone in HEK 293T cells, eGFP-NRF2 was mainly concentrated in the nucleus, while AsRed-TRIM16 was distributed diffusely throughout the cytoplasm. When eGFP-NRF2 and AsRed-TRIM16 were co-transfected, eGFP-NRF2 was mainly colocalized with AsRed-TRIM16 in the cytoplasm. The results of the confocal microscopy images demonstrate that TRIM16 could colocalize with NRF2 in the cytoplasm (Figure 5B). However, the interaction between NRF2 and TRIM16 is weakened in the state of highly pathogenic avian influenza virus infection (Figure 5C).

### 3.6. TRIM16 Regulates the SQSTM1-KEAP1-NRF2 System

Next, the molecular mechanism of TRIM16 in alleviating oxidative stress induced by influenza virus infection was explored. It is well known that SQSTM1/NRF2/KAEP1 plays an indispensable role in pathways that resist oxygen-induced stress. The mRNA and protein expression levels of SQSTM1, NRF2, and KEAP1 were determined. The results show that the mRNA and protein expression levels of SQSTM1 and NRF2 significantly increased after 24 h of transfection with HA-TRIM16 in A549 cells. Conversely, KEAP1 protein expression was significantly down-regulated, but KEAP1 mRNA levels had not obviously changed (Figure 6A–F). To explore whether a dose-dependent relationship between TRIM16 and SQSTM1, NRF2, and KEAP1 could be found, 1, 2, or 4 μg of HA-TRIM16 was transfected into A549 cells to determine SQSTM1, NRF2, or KEAP1 protein expression levels. The results show that SQSTM1 and NRF2 increased with TRIM16 in a dose-dependent manner (Figure 6G,H), while KEAP1 decreased with TRIM16 in a dose-dependent manner (Figure 6I). These results show that SQSTM1 and NRF2 were positively regulated by TRIM16 at protein and mRNA levels, while KEAP1 was negatively regulated by TRIM16 at the protein level.

The functions of the TRIM16 protein as part of the SQSTM1/NRF2/KEAP1 complex were verified. TRIM16-FLAG and HA-SQSTM1, HA-SQSTM1, and FLAG-NRF2 were co-transfected in HEK 293T cells for CoIP experiments. When the lysate was immunoprecipitated with anti-HA agarose beads, FLAG-NRF2 was co-immunoprecipitated with HA-SQSTM1 using anti-FLAG antibodies and TRIM16-FLAG was co-immunoprecipitated with HA-SQSTM1 using anti-FLAG antibodies (Figure 6J,K). These results suggest that TRIM16 formed complexes with SQSTM1 and NRF2 to undertake biological functions.

Under basal conditions, to determine the cause of TRIM16-mediated NRF2 stability, HA-KEAP1, eGFP-TRIM16, and FLAG-NRF2 were transfected into HEK 293T cells, and HA-KEAP1 and FLAG-NRF2 were transfected into cells as controls. The results indicate that TRIM16 displaced KEAP1 from NRF2 (Figure 6L). SQSTM1 interacted with KEAP1, sequestered it, and then mediated its autophagic degradation. To investigate whether TRIM16 causes an increase in the interaction between KEAP1 and SQSTM1, HA-KEAP1, FLAG-SQSTM1, and eGFP-TRIM16 were transfected into HEK293T cells. The data show that TRIM16 led to an increase in the interaction between KEAP1 and SQSTM1 in immunoprecipitation assays (Figure 6M). HA-SQSTM1, FLAG-NRF2, and eGFP-TRIM16 were then transfected into HEK 293T cells, and HA-SQSTM1 and FLAG-NRF2 were transfected into cells as controls. In the normal state of the cell, SQSTM1 has the function of stabilizing NRF2. Our results showed that TRIM16 enhanced the SQSTM1-NRF2 interaction (Figure 6N). TRIM16 stabilizes NRF2 by at least two mechanisms: (1) by substituting KEAP1 and the (2) increasing SQSTM1-NRF2 interactions. Collectively, the data shown here suggest that TRIM16 is an integral part of the SQSTM1/KEAP1/NRF2 system and positively regulates expression of SQSTM1 and NRF2.

### 3.7. TRIM16 Affects Ubiquitination Status of NRF2 

To investigate whether NRF2 was modified by TRIM16 ubiquitination in the normal state of the cell, the ubiquitination modification of NRF2 by the TRIM16 protein was detected using CoIP. HA-UB, FLAG-NRF2, and eGFP-TRIM16 were co-transfected into HEK293T cells. The results of CoIP showed that TRIM16 modified NRF2 via ubiquitination of the E3 ubiquitin ligase function (Figure 7A). To explore how TRIM16 affects NRF2 ubiquitination, an immunoprecipitation analysis was performed in the presence of two different ubiquitin protein variants, one ubiquitinated only at lysine at position 48 (HA-K48-UB) and the other only ubiquitinated at lysine at position 63 (HA-K63-UB). The results of the CoIP experiment showed that the polyubiquitination of K48-linked NRF2 was similar to that of ordinary polyubiquitination, but TRIM16 protein led to an increase in K63-linked poly-ubiquitination of NRF2 (Figure 7B,C). These results suggest that NRF2 is ubiquitinated by TRIM16 through an increase in the K63-linked polyubiquitination of NRF2.

## 4. Discussion

The concept of oxidative stress was first proposed on the basis of research on human aging [37]. The imbalance between oxidants and antioxidants when organisms are exposed to adverse stimuli is the widely accepted definition of oxidative stress [38]. On the one hand, viral infections trigger the production of certain ROS, which play an important role in inducing cell damage. On the other hand, excessive production of ROS leads infected cells to a chronic non-acute state of oxidative stress, which leads to an acceleration of disease progression. Large quantities of ROS and their resulting superoxides and derivatives are the main cause of lung damage caused by influenza virus infection. ROS can promote lung damage and inflammation caused by the influenza A virus infection [39,40]. However, the exact timing at which influenza viruses induce oxidative stress in host cells remains unclear. The molecular mechanisms underlying host cell resistance to oxidative stress induced by influenza virus infection have not been reported.

Viral infections lead to the production of a large number of ROS. However, the timing at which viruses induce ROS production in host cells does not appear to be consistent. DENV produced large amounts of ROS 48 h after infecting host cells [8]. However, in our study A549 cells produced large amounts of ROS within 24 h of infection with the H5N1 HPAIV. The discrepancy may be caused by infection with different viruses. The H5N1 HPAIV induced relatively low levels of ROS in A549 cell within 12 h of infection. This finding may be related to the fact that the virus only attached to the surface of the host cell and did not enter the cell [41].

Some viruses regulate oxidative stress through the modulation of the NRF2 antioxidant response pathway. For example, fever with thrombocytopenia syndrome virus (SFTSV) was shown to inhibit the activation of the NRF2 antioxidant signaling pathway to favor viral replication in the host [42]. Severe acute respiratory syndrome corona virus 2 (SARS-CoV-2) was found to lead to an increase in viral replication by causing an increase in the degree of oxidative stress in host cells through damage to the NRF2/Hmox1 axis by the non-structural viral protein NSP14 [43]. The NS2B3 complex of DENV damages NRF2 regulators to inhibit the antioxidant gene network, leading to increased ROS levels and promoting viral replication [8]. Influenza virus regulation of NRF2 activity has also been previously contradictorily demonstrated. On the one hand, the NRF2 antioxidant network was activated by influenza virus-induced oxidative stress to protect infected cells from damage [44]. On the other hand, proteome analysis found that the phosphorylated form of NRF2 did not enter the nucleus after infection with a highly pathogenic influenza virus [45], suggesting that influenza viruses lead to an inhibition of the activation of the NRF2 pathway. In another study, NRF2 knockout increased susceptibility to influenza A virus infection, while overexpression of NRF2 in alveolar epithelial cells caused a reduction in viral replication and associated oxidative stress [46]. The antioxidant n-acetyl cysteine (NAC) led to inhibition of the replication of H5N1 influenza viruses in A549 cells [47]. In this study, we found that NRF2 was increased by overexpression of TRIM16 to inhibit the replication of H5N1 subtype highly avian influenza virus in A549 cells.

The cystine/glutamate antiporter SLC7A11 (also commonly known as xCT), which plays functions in importing cystine for glutathione biosynthesis and antioxidant defense, was overexpressed in multiple human cancers. Activating transcription factor 4 (ATF4) was one of the main transcription factors that induce transcription of SLC7A11 [48]. Whether overexpressing TRIM16 affected ATF4 and caused a significant down-regulation of SLC7A11 in cells needs further studies to address this question. Since SLC7A11 is closely related to ferroptosis in cells, TRIM16 may also regulate the ferroptosis in cells.

Except for playing a crucial role in the biogenesis of protein aggregation [49], SQSTM1 regulates NRF2, which plays a key role in protein homeostasis and redox homeostasis [11,18,50,51,52]. Under normal circumstances, NRF2 forms a complex with KEAP1, a well-known negative regulator of NRF2 [53]. Since KEAP1 acts as an adaptor protein for cullin-3-based E3 ubiquitin ligase, this dimeric NRF2/KEAP1 complex enabled continuous ubiquitination and proteasome degradation of NRF2 [54]. Since SQSTM1 is regulated by autophagy, the cellular levels of SQSTM1 have been used as a marker to study autophagy flux. However, in a state of oxidative stress, autophagy activated by oxidative stress was found to cause depletion of SQSTM1 in HeLa cells and glioblastoma cell lines, independent of ubiquitin proteasome and transcription [55,56]. Therefore, the molecular mechanism by which TRIM16 alleviates virus-induced oxidative stress may also be involved in autophagy. In the present study, we found that the TRIM16 protein positively regulated the expression of SQSTM1 and NRF2 at the mRNA and protein levels and regulated the expression of SQSTM1 and NRF2 protein in a dose-dependent manner. TRIM16 also negatively regulated KEAP1 in a dose-dependent manner. However, whether TRIM16 was still regulated by knocking down or knocking out the NRF2 pathway is still unknown. Our data indicate that TRIM16 stabilized NRF2 under basal conditions primarily in two ways. On the one hand, TRIM16 led to an increase in the interaction of NRF2 with SQSTM1. On the other hand, TRIM16 caused a reduction in the interaction between NRF2 and KEAP1. Whether TRIM16 regulation of SQSTM1/NRF2/KEAP1 is related to the autophagy pathway needs further studies to verify such regulation. 

TRIM16 belongs to the TRIM family of proteins, and most members are RING domains containing E3 ligase [20]. TRIM16 contains a deleted RING domain, but it still has E3 ligase activity [26]. Ubiquitin precursor proteins in humans are encoded by the Ubb, Ubc, UBA52, and RPS27A genes, of which Ubb and Ubb have been shown to be up-regulated under oxidative and proteotoxic stress conditions [57]. K48-linked polyubiquitinated proteins were considered targets for proteasome degradation, while K63-linked polyubiquitination was important for multiple functions, including protein transport, protein oligomerization, protein stability, and activation of signaling pathways [58,59,60]. In this study, we found that NRF2 was ubiquitinated by TRIM16 through increasing K63-linked poly-ubiquitination. However, the relevant domains and key sites of TRIM16 ubiquitination of NRF2 need to be further explored.

## 5. Conclusions

Overexpression of the TRIM16 protein can reduce the titer of highly pathogenic avian influenza virus and promote the expression of antioxidant genes in A549 cells through the SQSTM1/NRF2/KEAP1 antioxidant pathway under H5N1 HPAIV infection conditions. Under basal conditions, TRIM16 interacts with NRF2 and can ubiquitinate NRF2 by causing an increase in K63-linked poly-ubiquitination of NRF2.

## Figures and Tables

**Figure 1 viruses-15-00391-f001:**
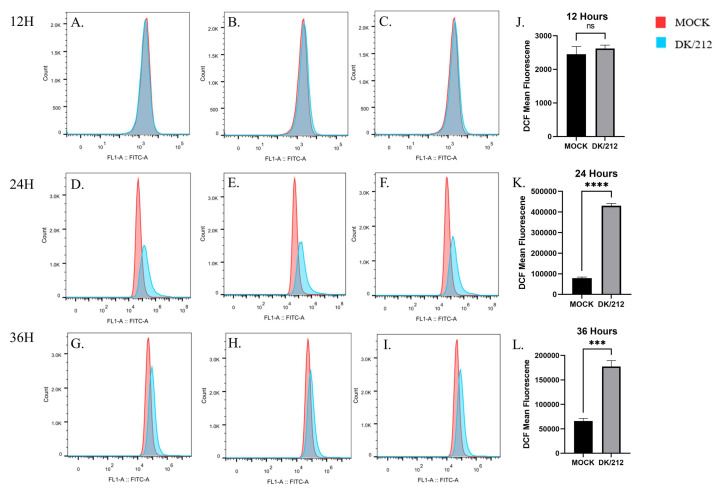
H5N1 subtype highly pathogenic avian influenza virus (DK/212) infection induced oxidative stress in A549 cells. (**A**–**C**,**J**) DK/212 virus infection and control in A549 cells at 12 hpi. (**D**–**F**,**K**) DK/212 virus infection and control in A549 cells at 24 hpi. (**G**–**I**,**L**) DK/212 virus infection and control in A549 cells at 36 hpi. Mean DCF fluorescence was determined via Flow cytometry as described in the methods. All data were expressed as means ± SD, the Student’s *t*-test was used to analyze the differences. (^ns^
*p* > 0.05, *** *p* < 0.001, **** *p* < 0.0001).

**Figure 2 viruses-15-00391-f002:**
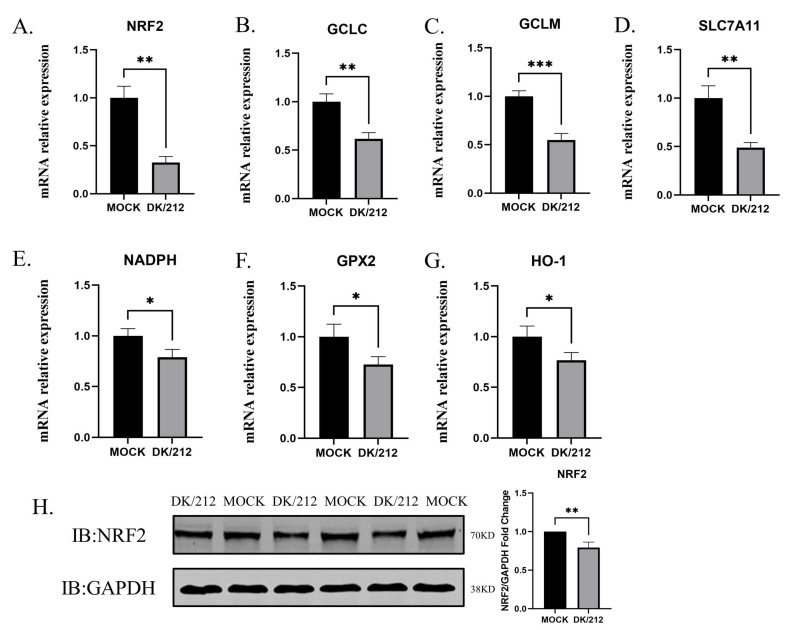
Relative expression levels of antioxidant genes in A549 cells at 24 hpi with H5N1 avian influenza virus. RNA was extracted from A549 cells at 24 hpi. The mRNA expression levels were analyzed by qPCR. GAPDH was used as loading control. The relative expression of (**A**) NRF2, (**B**) GCLC, (**C**) GCLM, (**D**) SLC7A11, (**E**) NADPH, (**F**) GPX2, (**G**) HO-1 were detected by real-time RT-PCR. (**H**) Changes in NRF2 protein expression in A549 cells 24 hpi. All data were expressed as means ± SD, the Student’s *t*-test was used to analyze the differences. (* *p* < 0.05, ** *p* < 0.01, and *** *p* < 0.001).

**Figure 3 viruses-15-00391-f003:**
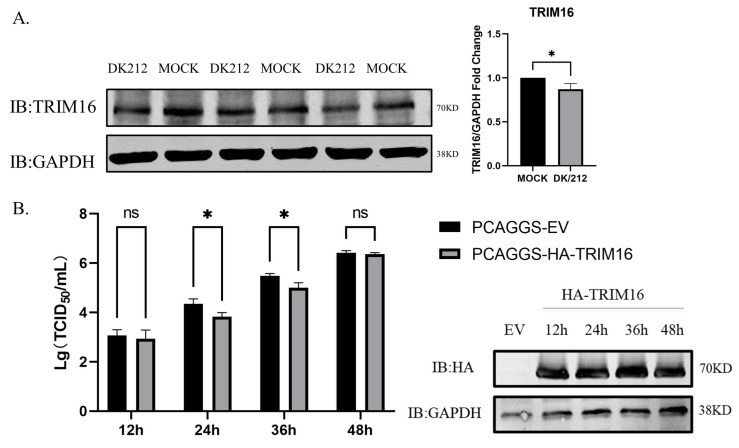
Overexpression of HA-TRIM16 reduces the titer of DK/212 virus in A549 cells. (**A**) Avian influenza virus infection reduces TRIM16 expression. Western blot analysis of TRIM16 protein expression in A549 cells infected with DK/212 virus for 24 h. GAPDH was used as loading control. (**B**) A549 cells were transfected with PCAGGS empty vector or PCAGGS-HA-TRIM16. At 24 h post transfection, the cells were infected with DK/212 at a dose of 0.1 MOI. The culture supernatants were harvested for viral titration at 12, 24, 36, and 48 hpi, respectively. All data were expressed as means ± SD, the Student’s *t*-test was used to analyze the differences (* *p* < 0.05).

**Figure 4 viruses-15-00391-f004:**
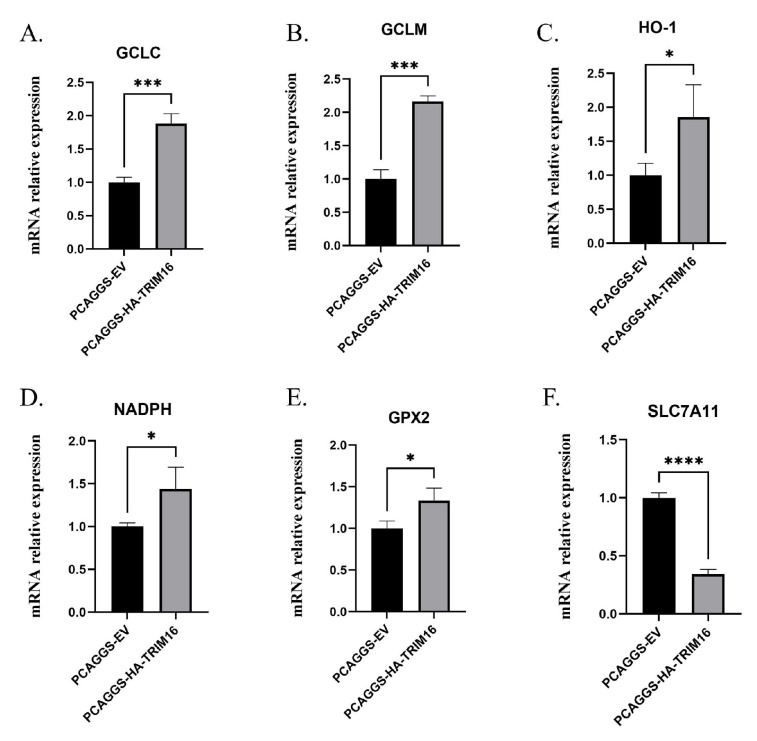
Relative expression of antioxidant genes after overexpression of TRIM16 in A549 cells for 24 h. At 24 hpi, RNA was extracted from A549. The mRNA expression levels were analyzed by qPCR. GAPDH was used as loading control. The relative expression of (**A**) GCLC, (**B**) GCLM, (**C**) HO-1, (**D**) NADPH, (**E**) GPX2, (**F**) SLC7A11 were detected by real-time RT-PCR. All data were expressed as means ± SD, the Student’s *t*-test was used to analyze the differences. (* *p* < 0.05, *** *p* < 0.001 and **** *p* < 0.0001).

**Figure 5 viruses-15-00391-f005:**
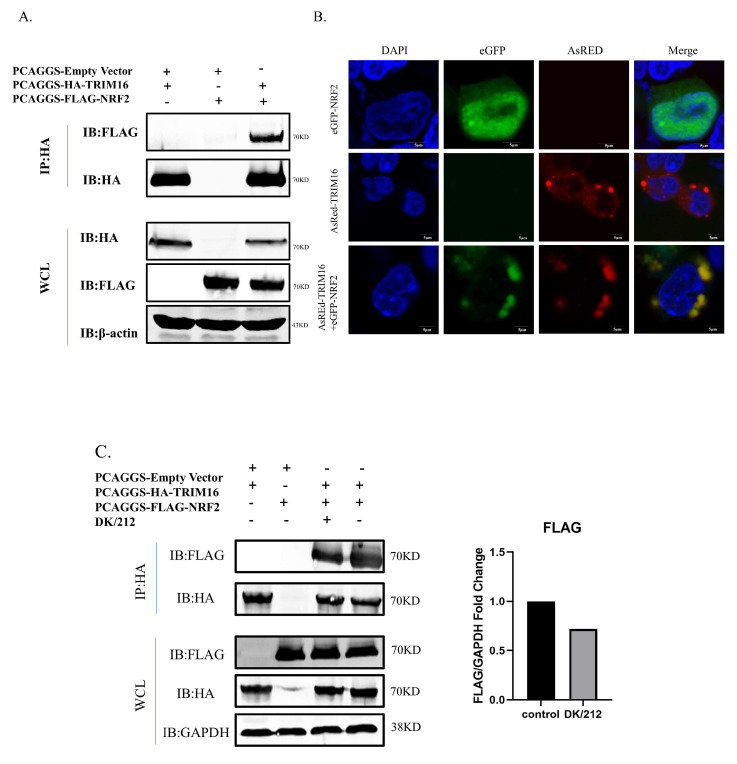
TRIM16 was shown to be an interaction protein of NRF2 and the two proteins were localized in the cytoplasm. (**A**) TRIM16 interacted with NRF2 in transfected cells. HEK 293T cells were transfected with PCAGGS-HA-TRIM16 and PCAGGS-FLAG-NRF2. After 24 h, the cells were lysed and immunoprecipitated using anti-HA agarose beads. The interaction was detected by Western blot using an anti-HA and anti-FLAG antibody. (**B**) TRIM16 and NRF2 were colocalized in transfected cells. HEK 293T cells were cultured on glass-bottom petri dishes until 50% confluence and were transfected with PCAGGS-AsRed-TRIM16 and PCAGGS-eGFP-NRF2. After 24 h, the transfected cells were fixed with cold methanol, stained with DAPI, and visualized by confocal laser scanning microscope. (**C**) After transfection of FLAG-NRF2 and HA-TRIM16 in 293T cells for 24 h, cells infected with DK/212 for 24 h.

**Figure 6 viruses-15-00391-f006:**
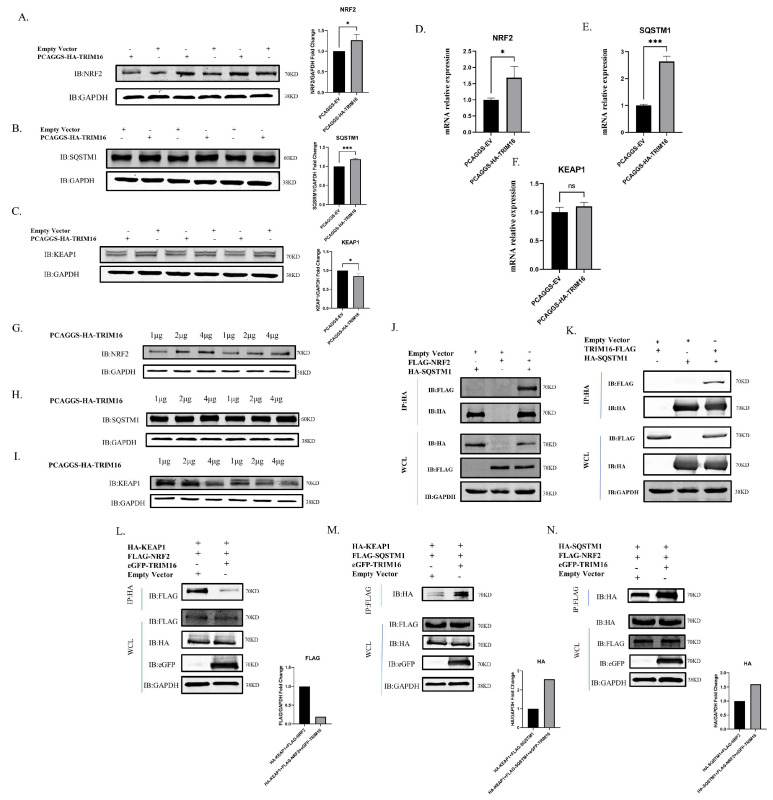
TRIM16 regulates the SQSTM1-KEAP1-NRF2 system. After 24 h of transfection of HA-TRIM16 detected the protein expression of (**A**) NRF2, (**B**) SQSTM1, and (**C**) KEAP1. After 24 h of transfection of HA-TRIM16 detected the mRNA expression of (**D**) NRF2, (**E**) SQSTM1, (**F**) and KEAP1, 1 μg, 2 μg, and 4 μg of TRIM16 were transfected in A549 cells to detect protein expression of (**G**) NRF2, (**H**) SQSTM1, and (**I**) KEAP1.GAPDH was used as a loading control. (**J**) SQSTM1 interacted with NRF2 in transfected cells. HEK 293T cells were transfected with PCAGGS-HA-SQSTM1 and PCAGGS-FLAG-NRF2. After 24 h, the cells were lysed and immunoprecipitated using anti-HA agarose beads. (**K**) TRIM16 interacted with SQSTM1 in transfected cells. HEK 293T cells were transfected with PCAGGS-HA-TRIM16 and PCAGGS-FLAG-SQSTM1. After 24 h, the cells were lysed and immunoprecipitated using anti-HA agarose beads. (**L**) Co-immunoprecipitation analysis of the interaction between NRF2 and KEAP1 in absence and presence of TRIM16. (**M**) Co-immunoprecipitation analysis of the interaction between KEAP1 and SQSTM1 in absence and presence of TRIM16. (**N**) Co-immunoprecipitation analysis of the interaction between NRF2 and SQSTM1 in absence and presence of TRIM16. All data were expressed as means ± SD, the Student’s *t*-test was used to analyze the differences. (^ns^
*p* > 0.05, * *p* < 0.05 and *** *p* < 0.001).

**Figure 7 viruses-15-00391-f007:**
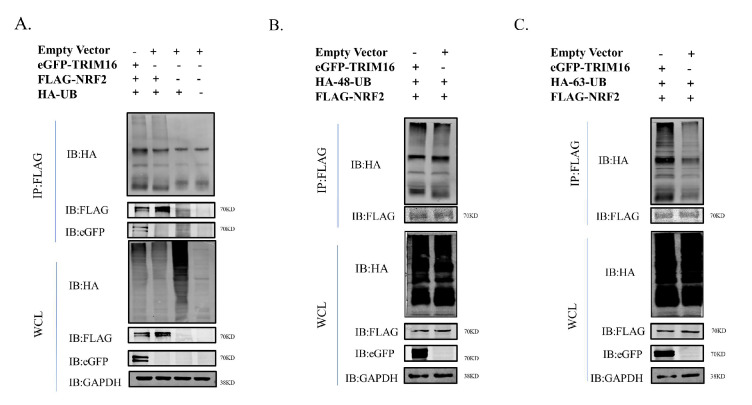
TRIM16 increases K63-linked poly-ubiquitination of NRF2. (**A**) HEK 293T cells were transfected with PCAGGS-FLAG-NRF2, HA-UB expression plasmids, PCAGGS-eGFP-TRIM16 or PCAGGS empty vector. (**B**) HEK 293T cells were transfected with PCAGGS-FLAG-NRF2, PCAGGS-eGFP-TRIM16 expression plasmids, HA-K48-UB or PCAGGS empty vector. (**C**) HEK 293T cells were transfected with PCAGGS-FLAG-NRF2, PCAGGS-eGFP-TRIM16 expression plasmids, HA-K63-UB or PCAGGS empty vector. GAPDH was used as loading control.

**Table 1 viruses-15-00391-t001:** Real-time quantitative PCR primers used in this study.

Gene	Forward Primer (5′-3′)	Reverse Primer (5′-3′)	GenBank Accession No.
GAPDH	GAAGGTGAAGGTCGGAGTCAAC	GAAGGTGAAGGTCGGAGTCAAC	NM_002046.7
NADPH	ACATATAGCATTGGGCACAC	CCAAATATTCTCCAGGCGTTT	NM_001256067.2
HO-1	CCAAATATTCTCCAGGCGTTT	CTTGGCCTCTTCTATCACCCT	NM_002133.2
GCLM	ACATGGCCTGTTCAGTCCT	ACATGGCCTGTTCAGTCCT	NM_002061.4
GCLC	ACAAGAAATATCCGACATAGGAG	ACAAGAAATATCCGACATAGGAG	NM_001498.3
GPX2	CATTGCCAAGTCCTTCTATGACC	CGAAGCCACATTCTCAATCAGC	NM_001256067.2
SLC7A11	TTTCTCATTAGCAGTTCCGAT	AGACGCAACATAGAATAACCTG	NM_014331.3
Nrf2	ACAACTCAGCACCTTATATCTCG	GGAACAAGGAAAACATTGCCAT	NM_006164.5
SQSTM1	CCCACGGCAGAATCAGCTT	GCTTCTTTTCCCTCCGTGCTC	NM_003900.5
KEAP1	CGTGGCTGTCCTCAATCGTCT	ATTGCTGTGATCATTCGCCACT	NM_203500.2

## Data Availability

The datasets presented in this study can be found in online repositories.

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
