# Peer review of "Overexpression of TRIM16 Reduces the Titer of H5N1 Highly Pathogenic Avian Influenza Virus and Promotes the Expression of Antioxidant Genes through Regulating the SQSTM1-NRF2-KEAP1 Axis"

_viruses, 2023, doi:10.3390/v15020391_

Round 1

Reviewer 1 Report

In the present research article, Liu et al describe the role of TRIM16 protein in reducing the oxidative stress induced by avian influenza virus in human cells. Tripartite motif (TRIM) proteins have been involved in immune response, cell proliferation, apoptosis and autophagy, among others. In this work, the authors show that TRIM16 may play a role in the reduction of oxidative stress-induced by H5N1 avian influenza virus DK/212 in human cells through the regulation of SQSTM1-NRF2-KEAP1 pathway.

In general, the paper is well written although correction of the english should be made.  The tittle of the work is not demonstrated in the work since it was not shown that TRIM16 overexpression alleviated/reduced oxidative stress in the context of H5N1 avian influenza virus DK/212 infection. Therefore, it is recommended to change the tittle.

The manuscript presented some problems of lack of consistency and information in the performance of the experiments, controls and missinterpretations that are indicated as follows.

The following revisions are needed

Figure 1, authors used MOI1 to show increased ROS levels during different times post infection concluding that 24 hpi is where most of the ROS production is observed. Therefore, decided to continue with that time point with following experiments. Further, the authors analyze the same time points addressed in figure 1 but with a different MOI 0,1 (figure 3). It is important that the results obtained are comparable under the same conditions otherwise, they are not correct.

Figure 3B, it is important to show as control, TRIM16 protein overexpression to understand how the system is working.

Figure 4, NFR2 mRNA level analysis is missing. It not depicted why it was not included.

Figure 5, in the context of TRIM16-NFR2 interactions (IFI, coIP) it is important to understand if the interaction between the proteins is altered in the context of infection.

Figure 6, G,H,I, a quantification graph of the bands should be added, in order to demonstrate the author’s conclusions (like the one shown in figure 6,A,B,C).

In Conclusions, sentence 457-458, there is a repetition of the tittle which is not demonstrated since reduction of ROS levels was not addressed in the context of virus infection and TRIM16 overexpression.

In many parts of the manuscript it is stated that virus replication is inhibited by TRIM16 overexpression, which is not correctly said. First, replication is specific of a part of the virus life cycle, the authors are addressing production of viral particles in the supernatant meaning that you are analyzing infection not only replication. Instead of inhibition, it is recommended to use the word reduced.

Minor

Sentence 92: First,

Sentence 108: Lipo2000

Sentence 402-404, Check sentence, is not clear

Reviewer 2 Report

this is a very interesting study. Text is well written and clear. Only minor points deserves to be improved. In particular: 

Lines 53-55: It deserves to be pointed out that NRF2/KEAP1 signalling is also involved in preventing cancer onset (PMID: 36335520) and inducing chemoresistance in cancer cells (PMID: 35901941). This is an important point to  add because highlight the multifaceted role of this signalling in physiological and pathological conditions.

Figures: When western blot images are shown, authors  must report the molecular weights

Figure 5b: Immunofluorescence images quality must be improved and scale bars must be visible 

Round 2

Reviewer 1 Report

The main concerns have been answered.